# Neural Networks with Structural Resistance to Adversarial Attacks

## Abstract

In adversarial attacks to machine-learning classifiers, small perturbations are added to input that is correctly classified. The perturbations yield adversarial examples, which are virtually indistinguishable from the unperturbed input, and yet are misclassified. In standard neural networks used for deep learning, attackers can craft adversarial examples from most input to cause a misclassification of their choice.

We introduce a new type of network units, called RBFI units, whose non-linear structure makes them inherently resistant to adversarial attacks. On permutation-invariant MNIST, in absence of adversarial attacks, networks using RBFI units match the performance of networks using sigmoid units, and are slightly below the accuracy of networks with ReLU units. When subjected to adversarial attacks based on projected gradient descent or fast gradient-sign methods, networks with RBFI units retain accuracies above 75%, while ReLU or Sigmoid see their accuracies reduced to below 1%. Further, RBFI networks trained on regular input either exceed or closely match the accuracy of sigmoid and ReLU network trained with the help of adversarial examples.

The non-linear structure of RBFI units makes them difficult to train using standard gradient descent. We show that RBFI networks of RBFI units can be efficiently trained to high accuracies using *pseudogradients,* computed using functions especially crafted to facilitate learning instead of their true derivatives.

## 1 Introduction

Machine learning via deep neural networks has been remarkably successful in a wide range of applications, from speech recognition to image classification and language processing. While very successful, deep neural networks are affected by adversarial examples: small, especially crafter modifications of correctly classified input that are misclassified (Szegedy et al. (2013)). The trouble with adversarial examples is twofold. The modifications to regular input are so small as to be difficult or impossible to detect for a human: this has been shown both in the case of images (Szegedy et al. (2013); Nguyen et al. (2015)) and sounds (Kurakin et al. (2016a); Carlini & Wagner (2018)). Further, the adversarial examples are in some measure transferable from one neural network to another (Goodfellow et al. (2014); Nguyen et al. (2015); Papernot et al. (2016a); Tramèr et al. (2017b)), so they can be crafted even without precise knowledge of the weights of the target neural network. At a fundamental level, it is hard to provide guarantees about the behavior of a deep neural network, when every correctly classified input is tightly encircled by very similar, yet misclassified, inputs.

Thus far, the approach for obtaining neural networks that are more resistant to adversarial attacks has been to feed to the networks, as training data, an appropriate mix of the original training data, and adversarial examples (Goodfellow et al. (2014); Madry et al. (2017)). In training neural networks using adversarial examples, if the examples are generated via efficient heuristics such as the *fast gradient sign method,* the networks learn to associate the specific adversarial examples to the original input from which they were derived, in a phenomenon known as *label leaking* (Kurakin et al. (2016b); Madry et al. (2017); Tramèr et al. (2017a)). This does not result in increased resistance to general adversarial attacks (Madry et al. (2017); Carlini & Wagner (2017b)). If the adversarial examples used in training are generated via more general optimization techniques, as in (Madry et al.

(2017)), networks with markedly increased resistance to adversarial attacks can be obtained, at the price of a more complex and computationally expensive training regime, and an increase in required network capacity.

We pursue here a different approach, proposing the use of neural network types that are, due to their structure, inherently impervious to adversarial attacks, even when trained on standard input only. In (Goodfellow et al. (2014)), the authors connect the presence of adversarial examples to the (local) linearity of neural networks. In a purely linear form $\sum_{i=1}^{n} x_i w_i$, we can perturb each $x_i$ by $\epsilon$, taking $x_i + \epsilon$ if $w_i > 0$, and $x_i - \epsilon$ if $w_i < 0$. This causes an output perturbation of magnitude $\epsilon \sum_{i=1}^{n} |w_i|$, or $n\bar{w}$ for $\bar{w}$ the average modulus of $w_i$. When the number of inputs $n$ is large, as is typical of deep neural networks, a small input perturbation can cause a large output change. Of course, deep neural networks are not globally linear, but the insight of (Goodfellow et al. (2014)) is that they may be sufficiently locally linear to allow adversarial attacks. Following this insight, we develop networks composed of units that are highly non-linear.

The networks on which we settled after much experimentation are a variant of the well known *radial basis functions* (RBFs) (Broomhead & Lowe (1988a); Chen et al. (1991); Orr (1996)); we call our variant RBFI units. RBFI units are similar to classical Gaussian RBFs, except for two differences that are crucial in obtaining both high network accuracy, and high resistance to attacks. First, rather than being radially symmetrical, RBFIs can scale each input component individually; in particular, they can be highly sensitive to some inputs while ignoring others. This gives an individual RBFI unit the ability to cover more of the input space than its symmetrical variants. Further, the distance of an input from the center of the Gaussian is measured not in the Euclidean, or $\ell_2$, norm, but in the infinity norm $\ell_\infty$, which is equal to the maximum of the differences of the individual components. This eliminates all multi-input linearity from the local behavior of a RBFI: at any point, the output depends on one input only; the $n$ in the above discussion is always 1 for RBFIs, so to say. The "I" in RBFI stands for the infinity norm.

Using deeply nonlinear models is hardly a new idea, but the challenge has been that such models are typically difficult to train. Indeed, we show that networks with RBFI units cannot be satisfactorily trained using gradient descent. To get around this, we show that the networks can be trained efficiently, and to high accuracy, using *pseudogradients.* A *pseudogradient* is computed just as an ordinary gradient, except that we artificially pretend that some functions have a derivative that is different from the true derivative, and especially crafted to facilitate training. In particular, we use pseudoderivatives for the exponential function, and for the maximum operator, that enter the definition of Gaussian RBFI units. Gaussians have very low derivative away from their center, which makes training difficult; our pseudoderivative artificially widens the region of detectable gradient around the Gaussian center. The maximum operator appearing in the infinity norm has non-zero derivative only for one of its inputs at a time; we adopt a pseudogradient that propagates back the gradient to all of its inputs, according to their proximity in value to the maximum input. Tampering with the gradient may seem unorthodox, but methods such as AdaDelta (Zeiler (2012)), and even gradient descent with momentum, cause training to take a trajectory that does not follow pure gradient descent. We simply go one step further, devising a scheme that operates at the granularity of the individual unit.

We show that with these two changes, RBFIs can be easily trained with standard random (pseudo)gradient descent methods, yielding networks that are both accurate, and resistant to attacks. To conduct our experiments, we have implemented RBFI networks on top of the PyTorch framework (Paszke et al. (2017)). The code will be made available in a final version of the paper. We consider *permutation invariant MNIST,* which is a version of MNIST in which the $28 \times 28$ pixel images are flattened into a one-dimensional vector of 784 values and fed as a feature vector to neural networks (Goodfellow et al. (2014)). On this test set, we show that for nets of 512,512,512,10 units, RBFI networks match the classification accuracy of networks of sigmoid units (($96.96 \pm 0.14$)% for RBFI vs. ($96.88 \pm 0.15$)% for sigmoid), and are close to the performance of network with ReLU units (($98.62 \pm 0.08$)%). When trained over standard training sets, RBFI networks retain accuracies over 75% for adversarial attacks that reduce the accuracy of ReLU and sigmoid networks to below 2% (worse than random). We show that RBFI networks trained on normal input are superior to ReLU and sigmoid networks trained even with adversarial examples. Our experimental results can be summarized as follows:

- In absence of adversarial attacks, RBFI networks match the accuracy of sigmoid networks, and are slightly lower in accuracy than ReLU networks.

- When networks are trained with regular input only, RBFI networks are markedly more resistant to adversarial attacks than sigmoid or ReLU networks.

- In presence of adversarial attacks, RBFI networks trained on regualar input provide higher accuracy than sigmoid or ReLU networks, even when the latter are trained also on adversarial examples, and even when the adversarial examples are obtained via general projected gradient descent (Madry et al. (2017)).

- RBFI networks can be successfully trained with pseudogradients; the training via standard gradient descent yields instead markedly inferior results.

- Appropriate regularization helps RBFI networks gain increased resistance to adversarial attacks.

Much work remains to be done, including experimenting with convolutional networks using RBFI units for images. However, the results seem promising, in that RBFI seem to offer a viable alternative to current adversarial training regimes in achieving robustness to adversarial attacks.

## 2  RELATED WORK

Adversarial examples were first noticed in Szegedy et al. (2013), where they were generated via the solution of general optimization problems. In Goodfellow et al. (2014), a connection was established between linearity and adversarial attacks. A fully linear form $\sum_{i=1}^{n} x_i w_i$ can be perturbed by using $x_i + \epsilon\,\text{sign}(w_i)$, generating an output change of magnitude $\epsilon \cdot \sum_{i=1}^{n} |w_i|$. In analogy, Goodfellow et al. (2014) introduced the *fast gradient sign method* (FGSM) method of creating adversarial perturbations, by taking $x_i + \epsilon \cdot \text{sign}(\nabla_i \mathcal{L})$, where $\nabla_i \mathcal{L}$ is the loss gradient with respect to input $i$. The work also showed how adversarial examples are often transferable across networks, and it asked the question of whether it would be possible to construct non-linear structures, perhaps inspired by RBFs, that are less linear and are more robust to adversarial attacks. This entire paper is essentially a long answer to the conjectures and suggestions expressed in Goodfellow et al. (2014).

It was later discovered that training on adversarial examples generated via FGSM does not confer strong resistance to attacks, as the network learns to associate the specific examples generated by FGSM to the original training examples in a phenomenon known as *label leaking* Kurakin et al. (2016b); Madry et al. (2017); Tramèr et al. (2017a). The FGSM method for generating adversarial examples was extended to an iterative method, I-FGSM, in Kurakin et al. (2016a). In Tramèr et al. (2017a), it is shown that using small random perturbations before applying FSGM enhances the robustness of the resulting network. The network trained in Tramèr et al. (2017a) using I-FSGM and ensemble method won the first round of the NIPS 2017 competition on defenses with respect to adversarial attacks.

Carlini and Wagner in a series of papers show that training regimes based on generating adversarial examples via simple heuristics, or combinations of these, in general fail to convey true resistance to attacks Carlini & Wagner (2017a;b). They further advocate measuring the resistance to attacks with respect to attacks found via more general optimization processes. In particular, FGSM and I-FGSM rely on the local gradient, and training techniques that break the association between the local gradient and the location of adversarial examples makes networks harder to attack via FGSM and I-FGSM, without making the networks harder to attack via general optimization techniques. In this paper, we follow this suggestion by using a general optimization method, projected gradient descent (PGD), to generate adversarial attacks and evaluate network robustness. Carlini & Wagner (2016; 2017b) also shows that the technique of *defensive distillation,* which consists in appropriately training a neural network on the output of another Papernot et al. (2016b), protects the networks from FGSM and I-FGSM attacks, but does not improve network resistance in the face of general adversarial attacks.

In Madry et al. (2017) it is shown that by training neural networks on adversarial examples generated via PGD, it is possible to obtain networks that are genuinely more resistant to adversarial examples. The price to pay is a more computationally intensive training, and an increase in the network capacity required. We provide an alternative way of reaching such resistance, one that does not rely on a new training regime.

## 3 RBFI Networks

In Goodfellow et al. (2014), the adversarial attacks are linked to the linearity of the models. Following this insight, we seek to use units that do not exhibit a marked linear behavior, and specifically, units which yield small output variations for small variations of their inputs measured in infinity norm . A linear form $g(\boldsymbol{x}) = \sum_i x_i w_i$ represents the norm-2 distance of the input vector $x$ to a hyperplane perpendicular to vector $\boldsymbol{w}$, scaled by $|\boldsymbol{w}|$ and its orientation. It is not advantageous to simply replace this norm-2 distance with an infinity-norm distance, as the infinity-norm distance between a point and a plane is not a very useful concept. It is preferable to consider the infinity-norm distance between points. Hence, we define our units as variants of the classical Gaussian *radial basis functions* (Broomhead & Lowe (1988b); Orr (1996)). We call our variant RBFI, to underline the fact that they are built using infinity norm.

An RBFI unit $\mathcal{U}(\boldsymbol{u}, \boldsymbol{w})$ for an input in $\mathbb{R}^n$ is parameterized by two vectors of weights $\boldsymbol{u} = \langle u_1, \ldots, u_n \rangle$ and $\boldsymbol{w} = \langle w_1, \ldots, w_n \rangle$ Given an input $\boldsymbol{x} \in \mathbb{R}^n$, the unit produces output

$$\mathcal{U}(\boldsymbol{u}, \boldsymbol{w})(\boldsymbol{x}) = \exp\left(-\|\boldsymbol{u} \odot (\boldsymbol{x} - \boldsymbol{w})\|_\infty^2\right) , \tag{1}$$

where $\odot$ is the Hadamard, or element-wise, product. In (1), the vector $\boldsymbol{w}$ is a point from which the distance to $\boldsymbol{x}$ is measured in infinity norm, and the vector $\boldsymbol{u}$ provides scaling factors for each coordinate. Without loss of expressiveness, we require the scaling factors to be non-negative, that is, $u_i \geq 0$ for all $1 \leq i \leq n$. The scaling factors provide the flexibility of disregarding some inputs $x_i$, by having $u_i \approx 0$, while emphasizing the influence of other inputs. Writing out (1) explicitly, we have:

$$\mathcal{U}(\boldsymbol{u}, \boldsymbol{w})(\boldsymbol{x}) = \exp\left(-\max_{1 \leq i \leq n}\left(u_i(x_i - w_i)\right)^2\right) . \tag{2}$$

The output of a RBFI unit is close to $1$ only when $\boldsymbol{x}$ is close to $\boldsymbol{w}$ in the coordinates that have large scaling factors. Thus, the unit is reminiscent of an And gate, with normal or complemented inputs, which outputs 1 only for one value of its inputs. Logic circuits are composed both of And and of Or gates. Thus, we introduce an Or RBFI unit by $\mathcal{U}^{\text{OR}}(\boldsymbol{u}, \boldsymbol{w}) = 1 - \mathcal{U}(\boldsymbol{u}, \boldsymbol{w})$. We construct neural networks out of RBFI units using layers consisting of And units, layers consisting of Or units, and mixed layers, in which the unit type is chosen at random at network initialization.

To form an intuitive idea of why networks with RBFI units might resist adversarial attacks, it is useful to compute the sensitivity of individual units to such attacks. For $x \in \mathbb{R}^n$ and $\epsilon > 0$, let $B_\epsilon(x) = \{x' \mid \|x - x'\|_\infty \leq \epsilon\}$ be the set of inputs within distance $\epsilon$ from $x$ in infinity norm. Given a function $f : \mathbb{R}^n \mapsto \mathbb{R}$, we call its *sensitivity to adversarial attacks* the quantity:

$$s = \sup_{x \in \mathbb{R}^n} \limsup_{\epsilon \to 0} \frac{\sup_{x' \in B_\epsilon(x)} |f(x) - f(x')|}{\epsilon} . \tag{3}$$

The sensitivity (3) represents the maximum change in output we can obtain via an input change within $\epsilon$ in infinity norm, as a multiple of $\epsilon$ itself. For a single ReLU unit with weight vector $\boldsymbol{w}$, the sensitivity is given by $s = \sum_{i=1}^n |w_i| = \|\boldsymbol{w}\|_1$. This formula can be understood by noting that the worst case for a ReLU unit corresponds to considering an input $\boldsymbol{x}$ for which the output is positive, and taking $x_i' = x_i + \epsilon$ if $w_i > 0$, and $x_i' = -\epsilon$ if $w_i < 0$ (Goodfellow et al. (2014)). Similarly, for a single sigmoid unit with weight vector $\boldsymbol{w}$, we have $s = \frac{1}{4}\|\boldsymbol{w}\|_1$, where the factor of $1/4$ corresponds to the maximum derivative of the sigmoid. For a RBFI unit $\mathcal{U}(\boldsymbol{u}, \boldsymbol{w})$, on the other hand, from (1) we have: $s = \frac{2}{e} \cdot \max_{1 \leq i \leq n} u_i^2 = \frac{2}{e} \cdot \|\boldsymbol{u}\|_\infty^2$. Thus, the sensitivity of ReLU and Sigmoid units increases linearly with input size, whereas the sensitivity of RBFI units is essentially constant with respect to input size. These formulas can be extended to bounds for whole networks. For a ReLU network with $K_0$ inputs and layers of $K_1, K_2, \ldots, K_M$ units, let $W^{(k)} = [w_{ij}]^{(k)}$ be its weight matrices, where $w_{ij}^{(k)}$ is the weight for input $i$ of unit $j$ of layer $k$, for $1 \leq k \leq K_M$. We can compute an upper bound $\hat{s}$ for the sensitivity of the network via:

$$\hat{\boldsymbol{s}}^{(0)} = \mathbf{1} , \qquad \hat{\boldsymbol{s}}^{(k)} = |W^{(k)}|\hat{\boldsymbol{s}}^{(k-1)} , \qquad \hat{s} = \|\hat{\boldsymbol{s}}^{(M)}\|_\infty . \tag{4}$$

The formula for Sigmoid networks is identical except for the $1/4$ factors. Using similar notation, for RBFI networks we have:

$$\hat{\boldsymbol{s}}^{(0)} = \mathbf{1} , \qquad \hat{s}_j^{(k)} = \frac{2}{e} \cdot \max_{1 \leq i \leq K_{k-1}} \hat{s}_i^{(k-1)}\left(u_{ij}^{(k)}\right)^2 , \qquad \hat{s} = \|\hat{\boldsymbol{s}}^{(M)}\|_\infty . \tag{5}$$

By connecting in a simple way the sensitivity to attacks to the network weights, these formulas suggest the possibility of using weight regularization to achieve robustness: by adding $c\hat{s}$ to the loss function for $c > 0$, we might be able to train networks that are both accurate and robust to attacks. We will show in Section 6.5 that such a regularization helps train more robust RBFI networks, but it does not help train more robust ReLU networks.

## 4  TRAINING RBFI NETWORKS VIA PSEUDOGRADIENTS

The non-linearities in (2) make neural networks containing RBFI units difficult to train using standard gradient descent, as we will show experimentally. The problem lies in the shape of Gaussian functions. Far from its peak for $x = w$, a function of the form (2) is rather flat, and its derivative may not be large enough to cause the vector of weights $w$ to move towards useful places in the input space during training. To obtain networks that are easy to train, we replace the derivatives for $\exp$ and $\max$ with alternate functions, which we call *pseudoderivatives*. These *pseudoderivatives* are then used in the chain-rule computation of the loss gradient in lieu of the true derivatives, yielding a *pseudogradient*.

**Exponential function.**  In computing the partial derivatives of (1) via the chain rule, the first step consists in computing $\frac{d}{dz}e^{-z}$, which is of course equal to $-e^{-z}$. The problem is that $-e^{-z}$ is very close to 0 when $z$ is large, and $z$ in (2) is $\|u \odot (x - w)\|_\infty^2$, which can be large. Hence, in the chain-rule computation of the gradient, we replace $-e^{-z}$ with the alternate "pseudoderivative" $-1/\sqrt{1 + z}$, which has a much longer tail.

**Max.**  The gradient of $y = \max_{1 \le i \le n} z_i$, of course, is given by $\frac{\partial y}{\partial z_i} = 1$ if $z_i = y$, and $\frac{\partial y}{\partial z_i} = 0$ if $z_i < y$. The problem is that this transmits feedback only to the largest input(s). This slows down training and can create instabilities. We use as pseudoderivative $e^{z_i - y}$, so that some of the feedback is transmitted to inputs $z_i$ that approach $y$.

One may be concerned that by using the loss pseudogradient as the basis of optimization, rather than the true loss gradient, we may converge to solutions where the pseudogradient is null, and yet, we are not at a minimum of the loss function. This can indeed happen. We experimented with switching to training with true gradients once the pseudogradients failed to yield improvements; this increased the accuracy on the training set, but barely improved it on the testing set. It is conceivable that more sophisticated ways of mixing training with regular and pseudo-gradients would allow training RBFI networks to higher accuracy on the testing set.

## 5  GENERATING ADVERSARIAL EXAMPLES

Given a correctly classified input $x$ for a network, and a perturbation size $\epsilon > 0$, an input $x'$ is an *adversarial example* for $\epsilon$ if $x'$ is misclassified, and $\|x - x'\|_\infty \le \eta$.

Consider a network trained with cost function $J(\theta, x, y)$, where $\theta$ is the set of network parameters, $x$ is the input, and $y$ is the output. Indicate with $\nabla_x J(\theta, x', y)$ the gradient of $J$ wrt its input $x$ computed at values $x'$ of the inputs, parameters $\theta$, and output $y$. For each input $x$ belonging to the testing set, given a perturbation amount $\epsilon > 0$, we produce adversarial examples $\tilde{x}$ with $\|x - \tilde{x}\|_\infty \le \epsilon$ using the following techniques.

**Fast Gradient Sign Method (FGSM)**  (Goodfellow et al. (2014)). If the cost were linear around $x$, the optimal $\epsilon$-max-norm perturbation of the input would be given by $\epsilon \, \text{sign}(\nabla_x J(\theta, x, y))$. This suggests taking as adversarial example:

$$\tilde{x} = [\![x + \epsilon \, \text{sign}(\nabla_x J(\theta, x, y))]\!]_0^1 \ , \tag{6}$$

where $[\![x]\!]_a^b$ is the result of clamping each component of $x$ to the range $[a, b]$; the clamping is necessary to generate a valid MNIST image.

**Iterated Fast Gradient Sign Method (I-FGSM)** (Kurakin et al. (2016a)). Instead of computing a single perturbation of size $\epsilon$ using the sign of the gradient, we apply $M$ perturbations of size $\epsilon/M$, each computed from the endpoint of the previous one. Precisely, the attack computes a sequence $\tilde{\boldsymbol{x}}_0, \tilde{\boldsymbol{x}}_1, \ldots, \tilde{\boldsymbol{x}}_M$, where $\tilde{\boldsymbol{x}}_0 = \boldsymbol{x}$, and where each $\tilde{\boldsymbol{x}}_{i+1}$ is obtained, for $0 \leq i < M$, by:

$$\tilde{\boldsymbol{x}}_{i+1} = \left\|\left[\tilde{\boldsymbol{x}}_i + \frac{\epsilon}{M}\operatorname{sign}(\nabla_{\boldsymbol{x}} J(\theta, \tilde{\boldsymbol{x}}_i, \boldsymbol{y}))\right]\right\|_0^1. \tag{7}$$

We then take $\tilde{\boldsymbol{x}} = \tilde{\boldsymbol{x}}_M$ as our adversarial example. This attack is more powerful than its single-step version, as the direction of the perturbation can better adapt to non-linear cost gradients in the neighborhood of $\boldsymbol{x}$ (Kurakin et al. (2016a)).

**Projected Gradient Descent (PGD)** (Madry et al. (2017)). For an input $\boldsymbol{x} \in \mathbb{R}^n$ and a given maximum perturbation size $\epsilon > 0$, we consider the set $B_\epsilon(\boldsymbol{x}) \cap [0, 1]^n$ of valid inputs around $\boldsymbol{x}$, and we perform projected gradient descent (PGD) in $B_\epsilon(\boldsymbol{x}) \cap [0, 1]^n$ of the negative loss with which the network has been trained (or, equivalently, projected gradient ascent wrt. the loss). By following the gradient in the direction of increasing loss, we aim at finding mis-classified inputs in $B_\epsilon(\boldsymbol{x}) \cap [0, 1]^n$. As the gradient is non-linear, to check for the existence of adversarial attacks we perform the descent multiple times, each time starting from a point of $B_\epsilon(\boldsymbol{x}) \cap [0, 1]^n$ chosen uniformly at random.

**Noise.** In addition to the above adversarial examples, we will study the robustness of our networks by feeding them inputs affected by noise. For a testing input $\boldsymbol{x}$ and a noise amount $\epsilon \in [0, 1]$, we produce an $\epsilon$-noisy version $\check{\boldsymbol{x}}$ via $\check{\boldsymbol{x}} = (1 - \epsilon)\boldsymbol{x} + \epsilon\boldsymbol{\chi}$, where $\boldsymbol{\chi}$ is a random element of the input space, which for MNIST is $[0, 1]^n$.

**Pseudogradient attacks: RBFI[psd].** We have implemented FGSM, I-FGSM, and PGD attacks for RBFI both relying on standard gradients, and relying on pseudogradients. In the results, we denote pseudogradient-based results via RBFI[psd]. The idea is that if pseudogradients are useful in training, they are likely to be useful also in attacking the networks, and an adversary may well rely on them.

Carlini & Wagner (2017b) show that many networks that resist FGSM and I-FGSM attacks can still be attacked by using general optimization-based methods. Thus, they and argue that the evaluation of attack resistance should include general optimization methods; the PGD attacks we consider are an example of such methods.

# 6 EXPERIMENTS ON PERMUTATION-INVARIANT MNIST

## 6.1 EXPERIMENTAL SETUP

**Implementation.** We implemented RBFI networks in the PyTorch framework (Paszke et al. (2017)). In order to extend PyTorch with a new function $f$, it is necessary to specify the function behavior $f(\boldsymbol{x})$, and the function gradient $\nabla_{\boldsymbol{x}} f$. To implement RBFI, we extend PyTorch with two new functions: a *LargeAttractorExp* function, with forward behavior $e^{-x}$ and backward gradient propagation according to $-1/\sqrt{1+x}$, and *SharedFeedbackMax,* with forward behavior $y = \max_{i=1}^n x_i$ and backward gradient propagation according to $e^{x_i - y}$. These two functions are used in the definition of RBFI units, as per (2), with the AutoGrad mechanism of PyTorch providing backward (pseudo)gradient propagation for the complete networks.

**Dataset.** We use the MNIST dataset (LeCun et al. (1998)) for our experiments, following the standard setup of 60,000 training examples and 10,000 testing examples. Each digit image was flattened to a one-dimensional feature vector of length $28 \times 28 = 784$, and fed to a fully-connected neural network; this is the so-called *permutation-invariant* MNIST.

**Neural networks.** We compared the accuracy of the following fully-connected network structures.

- **ReLU** networks (Nair & Hinton (2010)) whose output is fed into a softmax, and the network is trained via cross-entropy loss.

| Network | Accuracy | FGSM, $\epsilon$=0.3 | I-FGSM, $\epsilon$=0.3 | PGD, $\epsilon$=0.3 | Noise, $\epsilon$=0.3 |
|---|---|---|---|---|---|
| ReLU | $\mathbf{98.62 \pm 0.08}$ | $1.98 \pm 0.42$ | $0.06 \pm 0.06$ | $67.40$ | $79.36 \pm 2.60$ |
| Sigmoid | $96.88 \pm 0.15$ | $0.71 \pm 0.43$ | $0.11 \pm 0.11$ | $38.78$ | $56.57 \pm 2.28$ |
| RBFI | $96.96 \pm 0.14$ | $\mathbf{94.90 \pm 0.35}$ | $\mathbf{93.27 \pm 0.48}$ | $\mathbf{93.32}$ | $\mathbf{96.23 \pm 0.08}$ |
| RBFI[psd] | $96.96 \pm 0.14$ | $\mathbf{85.88 \pm 2.02}$ | $\mathbf{78.92 \pm 1.91}$ | $\mathbf{90.74}$ | $\mathbf{96.23 \pm 0.08}$ |

Table 1: Performance of 512-512-512-10 networks for MNIST testing input, and for input corrupted by adversarial attacks and noise computed with perturbation size $\epsilon = 0.3$.

- **Sigmoid** networks trained with square-error loss.
- **RBFI** networks, trained using square-error loss. For a RBFI network with $m$ layers, we denote its type as RBFI($K_1, \ldots, K_m \mid t_1, \ldots, t_m$), where $K_1, \ldots, K_m$ are the numbers of units in each layer, and where the units in layer $i$ are And units if $t_i = \wedge$, Or units if $t_i = \vee$, and are a random mix of And and Or units if $t_m = *$.

Square-error loss worked as well or better than other loss functions for Sigmoid and RBFI networks. Unless otherwise noted, we use networks with layers 512, 512, 512, and 10 units, and in case of RBFI networks, we used geometry RBFI($512, 512, 512, 10 \mid \wedge, \vee, \wedge, \vee$). For RBFI networks we use a bound of $[0.01, 3]$ for the components of the $u$-vectors, and of $[0, 1]$ for the $w$-vectors, the latter corresponding to the value range of MNIST pixels. We experimented with RBFI networks with various geometries, and we found the performance differences to be rather small, for reasons we do not yet fully understand. We trained all networks with the AdaDelta optimizer (Zeiler (2012)), which yielded good results for all networks considered.

**Attacks.** We applied FGSM, I-FGSM, and noise attacks to the whole test set. In I-FGSM attacks, we performed 10 iterations of (7). As PGD attacks are considerably more computationally intensive, we apply them to one run only, and we compute the performance under PGD attacks for the first 5,000 examples in the test set. For each input $x$ in the test set, we perform 20 searches, or restarts. In each search, we start from a random point in $B_\epsilon(x)$ and we perform 100 steps of projected gradient descent using the AdaDelta algorithm to tune step size; if at any step a misclassified example is generated, the attack is considered successful.

## 6.2 PERFORMANCE OF RBFI, RELU, AND SIGMOID NETWORKS

In Table 1 we summarize the results on the accuracy and resistance to adversarial examples for networks trained on the standard MNIST training set. The results are computed from 10 training runs for ReLU and Sigmoid networks, and from 5 runs for RBFI and RBFI[psd]. In each run we used different seeds for the random generator used for weight initialization; each run consisted of 30 training epochs. In a result of the form $a \pm e$, $a$ is the percentage accuracy, and $e$ is the standard deviation in the accuracy of the individual runs.

In absence of perturbations, RBFI networks lose $(1.66 \pm 0.21)\%$ performance compared to ReLU networks (from $(98.62 \pm 0.07)\%$ to $(96.96 \pm 0.14)\%$), and perform comparably to sigmoid networks (the difference is below the standard deviation of the results). When perturbations are present, in the form of adversarial attacks or noise, the performance of RBFI networks is superior.

We note that the FGSM and I-FGSM attacks performed using regular gradients are not effective against RBFI networks. This phenomenon is called *gradient masking:* the gradient in proximity of valid inputs offers little information about the possible location of adversarial examples Carlini & Wagner (2017b). Pseudogradients do avoid gradient masking, and indeed the most effective attack against RBFI networks is I-FGSM performed using pseudogradients, which lowers the accuracy to $(78.92 \pm 1.91)\%$ for $\epsilon = 0.3$.

## 6.3 PERFORMANCE OF NETWORKS TRAINED WITH ADVERSARIAL EXAMPLES

Including adversarial examples in the training set is the most common method used to make neural networks more resistant to adversarial attacks (Goodfellow et al. (2014); Madry et al. (2017)). We explored whether ReLU and Sigmoid networks trained via a mix of normal and adversarial exam-

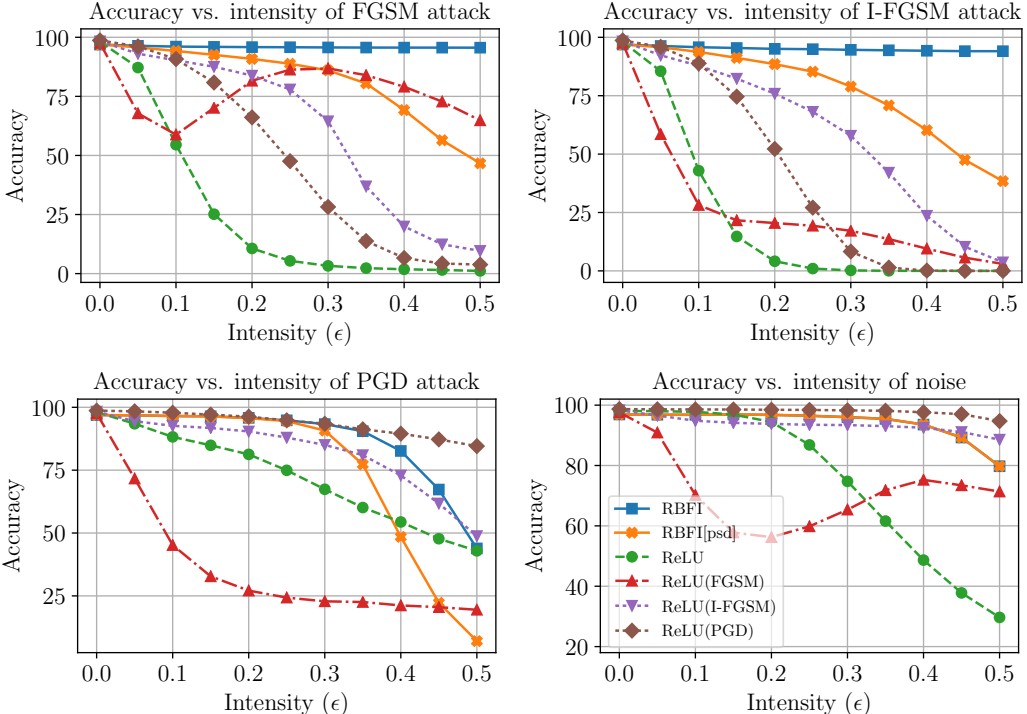

Figure 1: Performance of ReLU networks trained with adversarial examples, vs. performance of RBFI network trained normally, with respect to adversarial input and noise. The line legend in the bottom right applies to all plots.

ples offer a resistance to adversarial attacks compared to that offered by RBFI networks trained on standard examples only. For brevity, we omit the results for Sigmoid networks, as they were consistently inferior to those for ReLU networks. We compared the performance of a RBFI network with that of ReLU network trained normally (indicated simply by ReLU), and with ReLU networks trained as follows:

- **ReLU(FGSM)** and **ReLU(I-FSGM):** for each $(\boldsymbol{x}, t)$ in the training set, we construct an adversarial example $\tilde{\boldsymbol{x}}$ via (6) or (7), and we feed both $(\boldsymbol{x}, t)$ and $(\tilde{\boldsymbol{x}}, t)$ to the network for training.

- **ReLU(PGD):** for each $(\boldsymbol{x}, t)$ in the training set, we perform 100 steps of projected gradient descent from a point chosen at random in $B_\epsilon(\boldsymbol{x}) \cap [0, 1]^n$; denoting by $\boldsymbol{x}'$ the ending point of the projected gradient descent, we feed both $(\boldsymbol{x}, t)$ and $(\boldsymbol{x}', t)$ to the network for training.

We generated adversarial examples for training for $\epsilon = 0.3$, which is consistent with Madry et al. (2017). Due to the high computational cost of adversarial training (and in particular, PGD adversarial training), we performed one run, and performed the training of ReLU networks for 10 epochs, which seemed sufficient for their accuracy to plateau.

The results are given in Figure 1. Overall, the best networks may be the simple RBFI networks, trained without the use of adversarial examples: for each class of attack, they exhibit either the best performance, or they are very close in performance to the best performer; this is true for no other network type. For PGD attacks, the best performance is obtained by ReLU(PGD) networks trained on PGD attacks, but this may be simply due to gradient masking: note that ReLU(PGD) networks do not perform well with respect to I-FGSM attacks. We note that ReLU(FGSM) networks seem to learn that $\epsilon = 0.3$ FGSM attacks are likely, but they have not usefully generalized the lesson, for instance, to attacks of size $0.1$. The S-shaped performance curve of ReLU(FGSM) with respect to FGSM or noise is known as *label leaking:* the network learns to recognize the original input given its perturbed version (Kurakin et al. (2016b)).

### 6.4 PSEUDOGRADIENTS VS. STANDARD GRADIENTS

We compared the performance achieved by training RBFI networks with standard gradients, and with pseudogradients. After 30 epochs of training $\text{RBFI}(512, 512, 512, 10 \mid *, *, *, \vee)$ networks, pseudogradients yielded $(96.79 \pm 0.17)\%$ accuracy, while regular gradients only $(86.35 \pm 0.75)\%$. On smaller networks, that should be easier to train, the gap even widened: for $\text{RBFI}(128, 128, 10 \mid *, *, \vee)$ networks, it went from $(95.00 \pm 0.29)\%$ for pseudogradients to $(82.40 \pm 3.72)\%$ for regular gradients.

### 6.5 LEARNING WITH REGULARIZATION

In Section 3, we developed upper bounds for the sensitivity of ReLU and RBFI networks to adversarial attacks on the basis of network weights. It is reasonable to ask whether, using those upper bounds as weight regularizations, we might achieve robustness to adversarial attacks.

For ReLU networks, the answer is substantially negative. We experimented adding to the loss used to train the network a term $c\hat{s}$, for $c \geq 0$ and $\hat{s}$ as in (4). We experimented systematically for many values of $c$. Large values prevented the network from learning. Smaller values resulted in little additional robustness: for $\epsilon = 0.3$, simple FGSM attacks lowered the network accuracy to below 10%.

For RBFI networks, regularization did help. The choice of upper bound for the components of the $\boldsymbol{u}$-vector influences the resistance of the trained networks to adversarial examples, as can be seen from (5). In the experiments reported thus far, we used an upper bound of 3. One may ask: would RBFI networks perform as well, if a higher bound were used? The answer is yes, provided weight regularization is used in place of a tighter bound. If we raise the bound to 10, and use no regularization, the accuracy under PGD attacks with $\epsilon = 0.3$ drops from $93.32\%$ to $83.62\%$. By adding to the loss the regularization $c\hat{s}$, for $c = 0.0001$ and $\hat{s}$ as in (5), we can recover most of the lost accuracy, obtaining accuracy $89.38\%$ at $\epsilon = 0.3$.

## 7 CONCLUSIONS

In this paper, we have shown that non-linear structures such as RBFI can be efficiently trained using artificial, "pseudo" gradients, and can attain both high accuracy and high resistance to adversarial attacks.

Much work remains to be done. One obvious and necessary study is to build convolutional networks out of RBFI neurons, and measure their performance and resistance to adversarial attacks in image applications. Further, many powerful techniques are known for training traditional neural networks, such as dropout Srivastava et al. (2014). It is likely that the performance of RBFI networks can also be increased by devising appropriate training regimes. Lastly, RBFIs are just one of many conceivable highly nonlinear architectures. We experimented with several architectures, and our experience led us to RBFIs, but it is likely that other structures perform as well, or even better. Exploring the design space of trainable nonlinear structures is clearly an interesting endeavor.

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
