# OpenReview forum: "Neural Networks with Structural Resistance to Adversarial Attacks"
_ICLR.cc/2019/Conference_

### Official Review · AnonReviewer1 · 2018-11-02
**Interesting idea, but limited evaluation of effectiveness as a defense**

**Rating:** 7
**Confidence:** 4

**Review:**

Summary: The paper proposes a new architecture to defend against adversarial examples. The authors propose a network with new type of hidden units (RBFI units). They also provide a training algorithm to train such networks and evaluate the robustness of these models against different attacks in the literature.

Main concern: I think the idea  proposed here of using RBFI units is very interesting and intuitive. As pointed out in the paper, the RBFI units make it difficult to train networks using standard gradient descent, because the gradients can be uninformative. They propose a new training algorithm based on "pseudogradients" to mitigate this problem. However, while evaluating the model against attacks, only gradient based attacks are used (like PGD attack of Madry et al., or Carlini and Wagner). It's natural to expect that since the gradients are uninformative, these attacks might fail. However, what if we considered similar "pseudogradient" based attacks? In particular, just use the same training procedure formulation to attack (where instead of minimizing loss like in training, we maximize loss)?
I think this key experiment is missing in the paper and without this evaluation, it's hard to claim whether the models are more robust fundamentally, or it's just gradient masking.


Revision: After the authors revision, I change my score since they addressed my main complaint about results using pseudogradient attacks

---

### Official Review · AnonReviewer3 · 2018-11-02
**Experiments are not convincing**

**Rating:** 5
**Confidence:** 3

**Review:**

This paper proposes an infinity norm variant of the RBF as the activation function of neural networks. The authors demonstrate that the proposed unit is less sensitive to the out-liar generated by adversarial attacks, and the experimental results on MNIST confirmed the robustness of the proposed method against several gradient-based attacks.

Intuitively, the idea should work well against the features of adversarial examples which are far from the center of the cluster of "normal" features. However, the experiments are not convincing enough to show this point, and the entire method looks like a simple gradient mask technique. In my opinion, two types of experiments should be further considered:

1. Pseudo-gradient-based attacks. Since the networks are trained using Pseudo gradients, all the attacks utilized in this paper should be pseudo-gradient-based as well.

2. Black-Box attacks which do not rely on the information provided by gradients, such as transferable adversarial examples.

Furthermore, the robustness revealed on the "noise" attack is interesting, I wish the authors could provide an analysis of the effects on feature distributions using different types of attacks.

---

### Official Review · AnonReviewer2 · 2018-11-02
**An interesting idea, but needs more comprehensive/diverse evaluations**

**Rating:** 5
**Confidence:** 3

**Review:**

This paper introduces a new neural network layer for the purposes of defending against "white-box" adversarial attacks (in which the adversary is provided access to the neural network parameters). The new network unit and its activation function are constructed in such a way that the local gradient is sparse and therefore is difficult to exploit to add adversarial shifts to the input. To train the networks in the presence of a sparse gradient signal, the authors introduce a "pseudogradient", and optimize this proxy-gradient to optimize the parameters. This training procedure shows competitive performance (after training) on the permutation-invariant MNIST dataset versus other more standard network architectures, but is more robust to both adversarial attacks and random noise.

High-level comments:
- Using only a single dataset, and one on which the classification problem is rather easy, is cause for concern. I would need to see performance on another dataset, like CIFAR 10, to be more convinced that this is a general pipeline. In Sec 4, the authors mention that, using the pseudogradient, "one may be concerned that ... we may converge ... and yet, we are not at a minimum of the loss function". They claim that "in practice it does not seem to be a problem" on their experiments. This claim is a bit weak considering only a single, simple dataset was used for training. It is not obvious to me that this would succeed for more complex datasets.
- I would also like to see an additional set of adversarial attacks that are "RBFI-aware". A motivated attacker who is aware of this technique might replace the gradient in the adversarial attack with the pseudogradient instead; I expect such an attack would be effective. While problematic in general, I do not think this is necessarily an overall weakness of the paper (since we, the community, should be investigating methods like these to obfuscate the process of exploiting neural network models), but I would still like to see results showing the impact/performance of adversarial training over the pseudo-gradient. (I do not expect this will be very much effort.)
- What is the purpose of showing robustness of your network models to random noise? It is nice/interesting to see that your results are more robust to random noise, but what is the intuition for why your network performs better?

Wording and minor comments:
- The abstract is rather lengthy, but should probably contain somewhere a spelling-out of RBFI, since it informs the reader that the radial basis function (with infinity-norm) is the structure of the new network unit.
- Sec 4: "...indicate that pseudogradients work much better than regular gradients" :: Please be more clear that this is context specific "...than regular gradients for training RBFI networks".
- Sec. 4 :: Try to be consistent to how you specify "z" in this section, you alternate between the 'infinity-norm' definition and the 'max' definition from Eq. (2). Try to homogenize these.
- In general, the paper was well-proofed and well-written and was easy to read (high clarity).
- To my knowledge, this work is a rather unique foray into solving this problem (original).

Overall, I think this work is an interesting idea to address a rather important concern in the Deep Learning community. While the idea has merit, the small set of experiments in this paper is not sufficiently compelling for me to immediately recommend publication. With a bit more work put into exploring the performance of this method on other datasets, this paper could be made more complete. (Also, since I am aware that space is limited, some of the details on the adversarial attacks from other publications can probably be moved to an appendix.)

---

### Public Comment · (anonymous) · 2018-10-31
**Gradient masking?**

It looks like this network might just be masking gradients. Can you check what happens when you extend Figure 1 to eps=0.5? Accuracy should drop to at most 10% if the attack is not being broken by gradient masking.

---

> ### Author Response · Authors · 2018-10-31
> **There is some gradient masking in (I-)FGSM, less (or none?) in PGD**
>
> Quick summary:
>
> * There is gradient masking for FGSM and I-FGSM, hence we used also PGD as comparison.  PGD indeed provides more informative results than FGSM and I-FGSM.
>
> * PGD, especially when using pseudogradients, can find most adversarial examples (see below for data).  For epsilon = 0.3, it still supports the finding that the accuracy of RBFI is above 90%.
>
> Full answer:
>
> For FGSM and I-FGSM, the nets with RBFI units (let's call them RBFI nets) do present gradient masking.  Gradient masking means that the gradient in correspondence to input points is not very helpful in finding adversarial examples, and so experiments with FGSM and I-FGSM over-estimate accuracy.
>
> Since, as we noted in the paper, there is masking for FSGM and I-FSGM, we included also the results for PGD (projected gradient descent) with multiple restarts.  The restart points are chosen uniformly at random in the neighborhood of size epsilon of each input point.  Using PGD in this way to look for adversarial examples, in case of infinity norm, is what is done in Madry et al. (Towards Deep Learning Models Resistant to Adversarial Attacks) and advocated in a series of papers by Carlini and Wagner.   In a sense, if you want to look for adversarial examples in a region, the most general thing you can do is use a general optimization method such as PGD.
>
> So the question is, how good is PGD? Is it also affected by masking?
> To answer this, we conducted two additional experiments, using more than the 20 restarts used in the paper.  We considered the following values for epsilon:
>
> epsilon = 0.3 : for this epsilon, we believe RBFI does better than ReLU
> epsilon = 0.5 : for this epsilon, we should obtain 0% accuracy, since any pixel can be turned to middle gray.  However, since digits contain black/white, the middle-gray conversion is right at the extreme of the 0.5-neighborhood, so we includes also
> epsilon = 0.55 : for which the accuracy should be 0%.
>
> Using PGD with 500 random restarts, we find the following accuracy as function of the restarts:
>
> Epsilon = 0.3, Accuracy at 10 restarts  94.4%
> Epsilon = 0.3, Accuracy at 100 restarts 92.4%
> Epsilon = 0.3, Accuracy at 500 restarts 91.8%
>
> Epsilon = 0.5, Accuracy at 10 restarts  50.8%
> Epsilon = 0.5, Accuracy at 100 restarts 28.6%
> Epsilon = 0.5, Accuracy at 500 restarts 20.6%
>
> Epsilon = 0.55, Accuracy at 10 restarts  32.2%
> Epsilon = 0.55, Accuracy at 100 restarts 12.8%
> Epsilon = 0.55, Accuracy at 500 restarts  8.0%
>
> If we use pseudogradients in PGD, then PGD becomes even more effective at finding adversarial inputs:
>
> Epsilon = 0.3,  Accuracy at 10  restarts 91.0%
> Epsilon = 0.3,  Accuracy at 100 restarts 90.7%
>
> Epsilon = 0.5,  Accuracy at 10  restarts 6.4%
> Epsilon = 0.5,  Accuracy at 100 restarts 4.5%
>
> Epsilon = 0.55, Accuracy at 10  restarts 1.1%
> Epsilon = 0.55, Accuracy at 100 restarts 0.7%
>
> As we see, for epsilon = 0.5 and epsilon = 0.55, PGD especially using pseudogradients is able to find the vast majority of adversarial examples with 100 restarts. Since for epsilon = 0.3, the accuracy is above 90% still, we have a strong indication that the true accuracy of RBFI networks, in presence of adversarial attacks with epsilon = 0.3, is above about 90%.
>
> We agree that this data should be included, at least in summary form, in the paper.

---

### Author Response · Authors · 2018-11-12
**Revised the paper to include the results of attacks conducted using pseudogradients**

We have revised the paper, including the results of attacks against RBFI networks that are conducted using pseudogradients.
The more complete results still support the robustness claim for RBFI networks wrt adversarial attacks.

Indeed, we had experimented with pseudogradient-based attacks before submitting the paper, and we had then decided to omit the results.  Mainly, we thought that pseudogradients were an ad-hoc idea, and we thought that while it was acceptable to use such non-standard idea in training, it was best to keep to standard notions -- standard attacks, and true gradients -- for evaluation.

In view of the comments, we have now agree that including the results on attacks based on pseudogradients is of interest.  As mentioned, the paper now contains results for both gradient- and pseudogradient-based attacks against RBFI networks.

---

### Meta-Review · Area_Chair1 · 2018-12-17
**reject**

**Confidence:** 5
**Recommendation:** Reject

**Metareview:**

The paper presents a novel unit making the networks intrinsically more robust to gradient-based adversarial attacks. The authors have addressed some concerns of the reviewers (e.g. regarding pseudo-gradient attacks) but experimental section could benefit from a larger scale evaluation (e.g. Imagenet).